# Changes in Daily Life, Physical Activity, GAD, Depression, and Personal Hygiene of Adolescents in South Korea Due to the COVID-19

**DOI:** 10.3390/healthcare10101881

**Published:** 2022-09-27

**Authors:** Eun-Yeob Kim, Chilhwan Oh, Hwa-Jung Sung, Jaeyoung Kim

**Affiliations:** 1Research Institute for Skin Image, Korea University College of Medicine, Seoul 08308, Korea; 2Department of Dermatology, Wonkwang University Hospital, Wonkwang University School of Medicine, Iksan 54538, Korea; 3Department of Hematooncology, College of Medicine, Korea University, Seoul 02841, Korea; 4Core Research & Development Center, Korea University Ansan Hospital, Ansan 15355, Korea

**Keywords:** COVID-19, daily life, adolescent, personal hygiene

## Abstract

This study used raw data obtained from the Adolescents’ Health Behavior Survey by government-affiliated agency open data. A total of 109,796 students were sampled. A comparative analysis was performed based on the year 2020 and when the COVID-19 pandemic occurred, in which we analyzed changes in adolescents’ depression and panic anxiety perception and dietary habits, physical factors, physical activity, and handwashing habits. There was no weight control in the second year compared to the first year of the COVID-19 pandemic, and obesity also increased. The continuation of the non-contact situation caused by COVID-19 led to a worsening of subjective health awareness, and the experience of generalized anxiety disorder, depression, sadness and hopelessness, and suicidal thoughts and attempts increased. The frequency of washing hands with soap before meals and after using the toilet at school or at home was reduced. As a result of this analysis, we believe that there needs to be a system of support in place to address the academic gaps and deficiencies in learning caused by COVID-19, and that psychological and emotional support needs to be strengthened at this time, as well as the issues to be supported after the end of the non-contact situation.

## 1. Introduction

In December 2019, the SARS-CoV-2 coronavirus disease 2019 (COVID-19) outbreak began in Wuhan, China, spreading worldwide. According to a report by state agencies in 2020, confirmed cases of COVID-19 were 37 million, deaths exceeded 1 million, and as of 28th January 2021, the spread of COVID-19 has continued to increase [1,2,3]. The World Health Organization (WHO) declared an ‘International Public Health Emergency (PHIC)’ for the novel COVID-19 virus on 31 January 2020, and a pandemic on 11 March 2020 [2,3]. Unlike existing infectious diseases, COVID-19 has caused great social, economic, and cultural changes as the pandemic has been prolonged worldwide [4,5]. Since the first confirmed case in South Korea on 23 January 2020, COVID-19 has become prevalent and has affected the daily life of people of all age groups. COVID-19 has had significant physical, emotional, and psychological impact [6] on the daily lives of adolescents whose identity had not yet been established [6,7]. Seong (2020) found that there was a difference in risk perception by age when asked, ‘If you are infected with COVID-19, how serious would it be to your health?’ [8]. As such, various risk perceptions and behaviors have emerged due to the unprecedented changes brought on by COVID-19 [8]. Honarvar et al. [9] used the Health Belief Model (HBM, Rosenstock, 1974) [10] to study knowledge, behavior, and risk perception about COVID-19, and the knowledge and behavioral recognition intentions were low [11,12]; the lower the knowledge or education level on COVID-19, the lower the perceived severity of the risk [9,13,14,15]. As a result, COVID-19 knowledge impacts risk-prevention behavior [11,16]. Children and adolescents lack the ability to protect themselves compared to adults [17], making it difficult to actively respond to disasters such as COVID-19 [18]. Wang et al. found that COVID-19 had a negative impact on the mental health of children and adolescents [19]. Jung [6] also reported that the ratio of study time and excessive study time for children increased compared to that before COVID-19, and that children that showed excessive media use and less than the recommended exercise time showed higher anxiety about the future [20]. In addition, it is expected that there will be unencountered changes in the daily life and mental health of adolescents due to the COVID-19 pandemic.

Therefore, to overcome the limitations caused by COVID-19, this study statistically analyzes the changes in daily life and mental health of adolescents based on data from the Korea Centers for Disease Control and Prevention (KDCA) and preventive adolescent health behavior survey data. Our goal is to improve the collective understanding of adolescents’ psychological and behavioral responses. By analyzing the psychological, emotional, and physical changes of adolescents based on the situation in 2020 and 2021, it is necessary to identify the impact of new infectious diseases, such as COVID-19, on adolescents, and to seek appropriate support policies. In addition, this study can be used as basic data on the educational environment and policy decisions for future youth having to deal with new infectious diseases.

## 2. Materials and Methods

### 2.1. Study Setting, Designs and Sampling

This study used raw data from the 16th (2020) and 17th (2021) “Youth Health Behavior Survey” conducted by the Ministry of Education, the Ministry of Health and Welfare, and the Korea Disease Control and Prevention Agency (KDCPA) (Approval No. 117058) [21,22,23]. The “Youth Health Behavior Survey” is a national representative data survey conducted on about 60,000 students every year by stratifying 800 schools (400 middle and high schools each) based on the table frame of the Ministry of Education’s basic statistics survey. A total of 109,796 people sampled 54,948 students in 2020 (28,961 middle school students, 25,984 high school students) and 54,848 students in 2021 (30,015 middle school students, 24,833 high school students). In this study, a comparative analysis was performed based on the 16th (August 2020) and 17th (August 2021) surveys in the midst of the COVID-19 pandemic. We analyzed changes in adolescents’ depression and panic anxiety perception and dietary habits, physical factors, physical activity, and handwashing habits through survey questions. The 16th (2020) and 17th (2021) survey data were collected from August to November 2020 and 2021 due to COVID-19.

### 2.2. Data Variables

In our study, the variables analysed were as follows: (1) general characteristics, namely kidney (m), weight (kg), obesity (BMI), age, gender, weight control effort and smartphone usage time; (2) subjective health/body type perception, stress perception, experience of sadness/despair over the past year, and suicidal thoughts and experience; (3) in the past 7 days, eating habits, including breakfast (bread, rice or wheat flour, oatmeal, cereal, etc.), fruit intake, sweet drink intake, ion drink intake (Gatorade, Pocari Sweat, etc.), juice drink intake (Coco Farm, Juicy Cool, etc.), coffee drink intake (coffee mix, Let’s Bee, etc.), fast food intake (pizza, hamburger, chicken, etc.), nutritional education, daily water intake (including bottled water, carbonated water, barley tea, etc.), physical activity (high-intensity physical activity that increases heart rate and leads to shortness of breath or sweating), strength exercise, and muscle-strengthening exercise; (4) handwashing habits, including washing hands before eating at school and at home, after school and after going to the bathroom, and when returning home after going out; and (5) comparative analysis of perceptions pre-COVID-19 with physical activity, breakfast, drinking, smoking, and depression.

Obesity was defined as a body mass index (BMI) of 25, where BMI = weight (kg)/height (m)^2^. An obesity level of 25 was based on the WHO Asia-Pacific region and the Korean Obesity Association. To determine the level of adolescent generalized anxiety disorder (GAD), the response results of the GAD 7 scale developed by Spizer et al., used in the Youth Health Behavior Survey, were used. This consists of seven questions in the form of a four-point Likert scale (0 = not at all to 3 = very much) related to anxiety or anxiety experienced in the last two weeks. The higher the combined score, the higher the pan-anxiety level. It can be judged that 0–4 points show normal behavior, 5–9 points show mild anxiety, 10–14 points show moderate anxiety, and 15 points or more indicate severe anxiety. The GAD 7 scale results, which are frequently used as a primary screening discrimination tool for GAD in medical institutions, were judged to be suitable for estimating the overall level of GAD in domestic adolescents.

### 2.3. Data Analysis

For statistical analysis, a database was secondarily processed with the creation of an analysis database, and the chi-square test or Mann–Whitney analysis was performed according to technical statistics, frequency analysis, and characteristics of variables. The analysis software used IBM SPSS Statistics for Windows, version 25.0 (IBM Corp., Armonk, NY, USA), and the statistical significance level was set at *p* < 0.05.

## 3. Results

### 3.1. Changes in General Characteristics in the First and Second Years of the COVID-19 Outbreak

Table 1 shows the results of changes in general characteristics in the first and second years of the COVID-19 outbreak. Obesity in the first year of the outbreak was 18.1%, but obesity increased in the second year to 19.3% (*p* < 0.001). The percentage of those not trying to lose weight in the past month increased from 45.1% to 46.4%, and efforts to lose weight decreased from 35.1% to 33.9% (*p* < 0.001). Smartphone use increased weekly during the COVID-19 outbreak, from 96.4% to 96.9% (*p* < 0.001), and there was no change in weekend smartphone use (*p* = 0.983) (Table 1).

### 3.2. Emotional and Cognitive Changes in the First and Second Years of the COVID-19 Outbreak

The results of emotional and cognitive changes in the first and second years of the COVID-19 outbreak are as follows. Subjective health perception deteriorated from 3.89 ± 0.90 in the first year of the outbreak to 3.77 ± 0.91 in the second year (*p* < 0.001). Usual stress was 3.27 ± 0.96 in the second year compared to 3.17 ± 0.94 in the first year of the outbreak (*p* < 0.001). GAD experiences increased to 11.13 ± 4.50 in the second year from 10.91 ± 4.37 in the first year of the outbreak (*p* < 0.001). In the past year, experiences of sadness and despair increased from 25.2% during the first year of the outbreak to 26.8% in the second year (*p* < 0.001), suicidal thoughts increased from 10.9% to 12.7% (*p* < 0.001), suicide plans increased from 3.6% to 4.0%, and suicide attempts increased from 2.0% to 2.3% (*p* < 0.001) (Table 2).

### 3.3. Changes in Diet and Daily Life in the First and Second Years of the COVID-19 Outbreak

Changes in diet and daily life over the past week in the first and second years of the COVID-19 outbreak were as follows. The frequency of breakfast consumption increased in the second year when compared to that in the first year of the outbreak, while the number of meals eaten five times a week in the second year did not change, and the response to all other answers decreased (*p* < 0.001). The frequency of fruit intake decreased in the “not eating” category in the first year to the “more than three times a day” category in the second year, while there was no change in eating twice a day, and the response of other answers increased (*p* < 0.001). The carbonated drink answer “no drink,” increased, while “drink 1 or 2 times a day’ answers were unchanged, and the other answers decreased (*p* < 0.001). The sweet drink intake answers “no drink and 1–2 times a week” decreased (*p* < 0.001). Fast food intake decreased in the “not to eat” category in the second year compared to that in the first year, while the frequency of intake increased 1–2/3–4/5–6 times a week (*p* < 0.001). All the respondents who drank more than 1 cup of water every day increased their intake in the second year compared to that in the first year (*p* = 0.005). Compared to heart rate in the first year, in the second year, heart rate increased for more than 60 min a day, mild physical activity increased from 1 to 6 d a week (*p* < 0.001), the response to high-intensity physical activity (20 min or more) increased from 1 d to 5 d per week (*p* < 0.001), and the other answers decreased by 2 weeks (Table 3).

### 3.4. Changes in Personal Hygiene Habits Due to the COVID-19 Outbreak

Changes in personal hygiene habits due to the COVID-19 outbreak was observed. “Hand washing with soap before eating at school, in the past week” decreased from 88.3% to 84.6% (*p* < 0.001). Handwashing using soap after using the toilet at school decreased from 97.0% to 96.5% (*p* < 0.001) in the past week. Handwashing using soap before eating at home decreased from 93.3% to 91.5% in the past week (*p* < 0.001). Handwashing with soap after using the toilet at home decreased from 97.3% to 96.8% in the past week (*p* < 0.001). Handwashing using soap after returning home decreased in the past week from 97.6% to 97.1% (*p* < 0.001) (Table 4).

### 3.5. Changes in Daily Life Due to the COVID-19 Outbreak

The results in changes in daily life due to the COVID-19 outbreak can be seen in Table 5. Physical activity decreased compared to that before the COVID-19 outbreak (47.4%), not eating breakfast (72.6%), drinking (81.8%) and smoking (83.6%) remained unchanged, whereas depression (36.0%) increased.

## 4. Discussion

The results of comparing the first year (2020) and the second year (2021) of the COVID-19 outbreak among adolescents are as follows. Between the first and second years of the outbreak, obesity increased by 1.2%, the number of adolescents making no weight control efforts increased by 1.3%, and efforts to lose weight decreased by 1.2%. Weekday smartphone usage time also increased by 0.5%. Subjective health conditions deteriorated in the second year compared to that in the first year of the outbreak. Stress and general anxiety disorder (GAD) were 0.10 and 0.22% higher in the second year when compared to that in the first year, respectively. “Experience of sadness and despair of the past year” increased by 1.6% from the first year, “suicide thoughts” by 1.8%, “suicide plans” by 0.4%, and “suicide attempt” by 0.3%.

Handwashing before eating at school decreased by 3.7%, handwashing after using the bathroom at school by 0.5%, handwashing before eating at home by 1.8%, handwashing after using the bathroom at home by 0.5%, and handwashing after returning home by 0.5%. Compared to before the outbreak, physical activity decreased by 79.8%. A total of 72.6% participants said they did not eat breakfast. Depression increased by 36.0%. Jung et al. [6] found that adolescents’ stress levels decreased slightly to 0.11 on average in 2020, and that the proportion of experiences of sadness/despair and suicidal thoughts experienced by adolescents decreased compared to that in 2019 [20]. However, in this study, adolescents in 2021 showed a 0.10 stress level, and experiences of sadness/desperation and suicidal thoughts increased. In addition, it was found that adolescents from poor families had a higher experience of sadness/despair than those from non-poor families, and that they had higher sadness/despair in 2021, the second year of the outbreak, than in 2020, the first year of the outbreak [20,24].

Kang et al. reported that obesity increased before the pandemic, and this result also showed that obesity was higher in 2021 than in 2020 [25]. Kang et al. reported that the frequency of eating breakfast, fast food intake, fruit intake, and carbonated drinks increased in the obese group before the COVID-19 outbreak [25], while the frequency of eating fruit, sweet drinks, and fast foods increased in 2021 when compared to that in 2020. It cannot be concluded that an increased frequency of intake has a negative effect on the body, such as obesity, and it is necessary to consider the amount and calories consumed. In the medical guidance, it was said that protecting ourselves from infection means washing our hands [26], and that personal hygiene is the best way to protect ourselves from infectious diseases. However, in this study, when comparing the first year to the second year, the personal hygiene of handwashing was not well-actioned. Of course, if a new infectious disease such as Middle-Eastern Respiratory Syndrome (MERS) appears, it may be useless [26], but handwashing in daily life is a very good way to defend oneself from germs and viruses. Kim et al. also stated that handwashing promotion activities had a significant effect on reducing the incidence of hospital infections [27], but that we should now practice handwashing in our daily life in order to defend against bacteria and viruses. Handwashing is the easiest form of personal hygiene to carry out in everyday life, but as an important preventive habit [27], daily handwashing habits should be made a part of daily life to prevent new infectious diseases such as COVID-19. This study also showed a tendency to decline in handwashing over time, supplementing education and promotion programs to ensure that a culture of proper hand hygiene practice is established.

Due to COVID-19, face-to-face classes such as schools, online remote classes, and outside activities have increased the time spent at home, and adolescents need to engage in regular physical activities, proper eating habits, and personal hygiene habits, but COVID-19 has had a negative impact on many of these areas [24,26]. Cho et al. found that the increase in online learning time after COVID-19 has led to an increase in smartphone use time and, as a result, the problem of smartphone overdependence [28], and in addition, for some groups of adolescents (such as multicultural family youth, and other family youth), the COVID-19 situation may reduce their experiences of sadness/despair, suicidal thoughts, and violence, while for others (such as adolescents from impoverished families and youth living in military areas), the COVID-19 situation may lead to higher experiences of sadness/despair or violence. Because the COVID-19 pandemic has increased the frequency of online classes and staying at home, it tends to cause obesity, adaptation disorders, depression, increased cognitive impulsivity, lack of behavioral control, and increased smartphone dependence [25], showing the same results as previous studies. As such, a new epidemic like COVID-19 has its own disease-causing impact, but because the links are intertwined in so many areas, new alternatives are needed to address the factors that pose a risk to adolescents while addressing the prevention of infectious diseases. The COVID-19 pandemic was an unprecedentedly difficult time for everyone around the world, but also caused difficulties for adolescents studying in school, academia, etc., and caused their daily life to be restricted, marked by confusion and difficulties. Society and the nation should aim to reduce the anxiety and discomfort caused by diseases such as COVID-19 to enable proper study.

## 5. Conclusions

The outbreak of COVID-19 increased the time that adolescents spent at home, and the frequency of breakfast, soda, sweet drinks, and water intake increased within their dietary habits. In addition, the frequency of muscle strengthening exercises and middle- and high-intensity physical activities increased. On the other hand, handwashing habits showed a trend of decreasing overall at school and home. Overall, compared to before the COVID-19 outbreak, the amount of physical activity decreased, and depression showed an increasing trend.

As a result of this analysis, we believe there needs to be a system of support to address the academic gaps and deficiencies in learning caused by COVID-19. Psychological and emotional support needs to be strengthened at this time, as well as in the issues to be supported after the end of a non-contact situation. In addition to creating psychological and emotional support policies to help disadvantaged youth during the COVID-19 pandemic, we need to consider what form of psychological and emotional support policy will be introduced for disadvantaged youth after the COVID-19 non-contact situation.

Since this is a cross-sectional study, there is a limitation in clearly grasping temporal pre-post relationships. However, we think that the online youth health behavior survey is meaningful because it is a representative survey of approximately 60,000 teenagers nationwide every year.

## Figures and Tables

**Table 1 healthcare-10-01881-t001:** Changes in general characteristics in the first and second years of the COVID-19 outbreak.

Characteristics		COVID-19				*X*^2^/z	*p-*Value
		2020		2021			
		M	SD	M	SD		
Height (m) ^4^		166.09	8.41	166.16	8.36	−1.308	0.191
Weight (kg) ^4^		59.79	13.20	60.11	13.73	−1.774	0.076
Body Mass Index ^4^		21.53	3.66	21.62	3.83	−1.611	0.107
Age ^4^		15.10	1.75	15.09	1.74	−0.712	0.476
Obesity ^3^	Normal	41,164	81.9	40,624	80.7	25.548	0.000
	Obesity	9093	18.1	9738	19.3		
Sex ^3^	Man	28,353	51.6	28,401	51.8	0.362	0.547
	Woman	26,595	48.4	26,447	48.2		
Weight loss efforts in the past month ^3^	No effort	24,774	45.1	25,436	46.4	31.397	0.000
	Tried to lose weight	19,272	35.1	18,590	33.9		
	Tried to gain weight	4220	7.7	3962	7.2		
	Maintained weight	6682	12.2	6860	12.5		
Smartphone use during the week ^3^	Did not use	1970	3.6	1725	3.1	16.352	0.000
	Used	52,978	96.4	53,123	96.9		
Weekend smartphone use ^3^	Did not use	1800	3.3	1798	3.3	0.000	0.983
	Used	53,148	96.7	53,050	96.7		

Average (M), Standard deviation (SD), ^3^ chi-square test (X^2^) (n = sample), ^4^ Mann–Whitney test (z), *p*-value < 0.05.

**Table 2 healthcare-10-01881-t002:** Emotional and cognitive changes in the first and second years of the COVID-19 outbreak.

Variable		COVID-19				Z ^3^	*p*-Value
		2020		2021			
		M ^1^	SD ^2^	M	SD		
Health perception		3.89	0.90	3.77	0.91	−23.036	0.000
Subjective body shape perception		3.18	0.97	3.17	0.98	−2.910	0.004
Stress perception		3.17	0.94	3.27	0.96	−18.228	0.000
Degree of fatigue recovery from sleep ^4^		2.94	1.14	2.72	1.10	−32.860	0.000
Feeling agitated, anxious, or irritable ^5^		1.56	0.77	1.61	0.79	−10.778	0.000
Inability to stop worrying ^5^		1.60	0.82	1.64	0.85	−7.299	0.000
Worrying too much about various things ^5^		1.97	0.95	2.00	0.97	−4.966	0.000
General unease ^5^		1.46	0.76	1.49	0.79	−7.553	0.000
Restlessness ^5^		1.27	0.62	1.30	0.65	−4.871	0.000
Easily irritable ^5^		1.70	0.87	1.73	0.88	−5.492	0.000
Feeling afraid that something terrible is about to happen ^5^		1.36	0.71	1.37	0.73	−3.171	0.002
GAD ^5^		10.91	4.37	11.13	4.50	−8.789	0.000
Sadness and despair ^6^	No	41,108	74.8	40,156	73.2	36.503	0.000
	Yes	13,840	25.2	14,692	26.8		
Suicidal thoughts ^6^	No	48,969	89.1	47,892	87.3	85.678	0.000
	Yes	5979	10.9	6956	12.7		
Suicide plan ^6^	No	52,995	96.4	52,642	96.0	16.479	0.000
	Yes	1953	3.6	2206	4.0		
Suicide attempt ^6^	No	53,827	98.0	53,603	97.7	6.875	0.009
	Yes	1121	2.0	1245	2.3		

^1^ M; average, ^2^ SD; standard deviation, ^3^ z; Mann–Whitney test, ^4^; in the 1st week, ^5^; in the past 2 weeks, ^6^; in the last year, *p*-value < 0.05.

**Table 3 healthcare-10-01881-t003:** Changes in diet and daily life in the first and second years of the COVID-19 outbreak.

Variable		COVID-19				*X* ^2 3^	*p-*Value
		2020		2021			
		N ^1^	% ^2^	N	%		
Breakfast (bread, sun food or wheat flour, oatmeal, cereal, etc.) ^4^	0th time	11,441	20.8	11,904	21.7	39.341	0.000
	1st time	4234	7.7	4031	7.3		
	2nd time	5010	9.1	4875	8.9		
	3rd time	4698	8.5	4527	8.3		
	4th time	3901	7.1	3825	7.0		
	5th time	6069	11.0	6494	11.8		
	6th time	3857	7.0	3853	7.0		
	7th time	15,738	28.6	15,339	28.0		
Fruit (fruit juice excluded) ^4^	0th time	7130	13.0	6544	11.9	56.939	0.000
	1~2nd time/1 week	17,643	32.1	17,639	32.2		
	3~4th time/1 week	14,613	26.6	15,293	27.9		
	5~6th time/1 week	5561	10.1	5586	10.2		
	1st time/everyday	5762	10.5	5800	10.6		
	2nd time/everyday	2583	4.7	2553	4.7		
	3rd time over/everyday	1656	3.0	1433	2.6		
Carbonated drinks ^4^	0th time	12,251	22.3	13,169	24.0	61.798	0.000
	1~2nd time/1 week	23,139	42.1	22,596	41.2		
	3~4th time/1 week	12,133	22.1	11,973	21.8		
	5~6th time/1 week	3677	6.7	3440	6.3		
	1st time/everyday	1958	3.6	1986	3.6		
	2nd time/everyday	861	1.6	900	1.6		
	3rd time over/everyday	929	1.7	784	1.4		
Sweet drinks (ion drinks, juice drinks, and coffee drinks excluded) ^4^	0th time	9205	16.8	8476	15.5	68.234	0.000
	1~2nd time/1 week	20,476	37.3	19,949	36.4		
	3~4th time/1 week	14,048	25.6	14,782	27.0		
	5~6th time/1 week	5498	10.0	5671	10.3		
	1st time/everyday	3430	6.2	3622	6.6		
	2nd time/everyday	1285	2.3	1386	2.5		
	3rd time over/everyday	1006	1.8	962	1.8		
Fast food (pizza, hamburger, chicken, etc.) ^4^	0th time	10,037	18.3	9319	17.0	43.112	0.000
	1~2nd time/1 week	31,255	56.9	31,284	57.0		
	3~4th time/1 week	10,861	19.8	11,294	20.6		
	5~6th time/1 week	1803	3.3	1975	3.6		
	1st time/everyday	656	1.2	639	1.2		
	2nd time/everyday	155	0.3	158	0.3		
	3rd time over/everyday	181	0.3	179	0.3		
Nutritional and eating habit education ^5^	No	28,653	52.1	31,518	57.5	313.477	0.000
	Yes	26,295	47.9	23,330	42.5		
Daily water (including bottled water, carbonated water, barley tea, etc.)	1 cup below per day	1966	3.6	1963	3.6	15.034	0.005
	2 cups per day	9583	17.4	9764	17.8		
	3 cups per day	11,910	21.7	12,222	22.3		
	4 cups per day	9790	17.8	9831	17.9		
	5 cups or more per day	21,699	39.5	21,068	38.4		
Physical activity causing shortness of breath for a total of 60 min over per day ^4^	0th time	21,111	38.4	18,250	33.3	409.631	0.000
	1st time/1 week	8432	15.3	8481	15.5		
	2nd time/1 week	7698	14.0	8790	16.0		
	3rd time/1 week	6328	11.5	7170	13.1		
	4th time/1 week	3248	5.9	3806	6.9		
	5th time/1 week	3517	6.4	3783	6.9		
	6th time/1 week	1185	2.2	1377	2.5		
	7th time/1 week	3429	6.2	3191	5.8		
High-intensity physical activity for 20 min causing shortness of breath or sweating ^4^	0th time	20,937	38.1	17,781	32.4	452.968	0.000
	1st time/1 week	10,185	18.5	10,510	19.2		
	2nd time/1 week	8159	14.8	9484	17.3		
	3rd time/1 week	5657	10.3	6511	11.9		
	4th time/1 week	2783	5.1	3208	5.8		
	5th time over/1 week	7227	13.2	7354	13.4		
Exercise to increase muscle strength such as push-ups, sit-ups, lifting weights, dumbbells, iron bars, and parallel bars (muscle strengthening exercises) ^4^	0th time	27,585	50.2	28,370	51.7	69.543	0.000
	1st time/1 week	8283	15.1	8112	14.8		
	2nd time/1 week	5539	10.1	5771	10.5		
	3rd time/1 week	4414	8.0	4337	7.9		
	4th time/1 week	2349	4.3	2282	4.2		
	5th time over/1 week	6778	12.3	5976	10.9		

^1^ N; sample, ^2^ %; percentage, ^3^ X^2^; chi-square test, ^4^; in the past week, ^5^; nutritional education in the past year, *p*-value < 0.05.

**Table 4 healthcare-10-01881-t004:** Changes in personal hygiene habits due to the COVID-19 outbreak.

Variable		COVID-19				*X* ^2 3^	*p*-Value
		2020		2021			
		N	%	N	%		
Washing hands using soap before eating at school *	Always	17,759	32.3	14,342	26.1	766.537	0.000
	Often	16,071	29.2	15,436	28.1		
	Sometimes	14,700	26.8	16,660	30.4		
	Not at all	6418	11.7	8410	15.3		
Hand washing using soap after using the toilet at school *	Always	37,007	67.3	36,467	66.5	46.631	0.000
	Often	11,839	21.5	11,617	21.2		
	Sometimes	4482	8.2	4831	8.8		
	Not at all	1620	2.9	1933	3.5		
Hand washing using soap before eating at home *	Always	21,931	39.9	20,789	37.9	156.279	0.000
	Often	16,903	30.8	16,370	29.8		
	Sometimes	12,400	22.6	13,061	23.8		
	Not at all	3714	6.8	4628	8.4		
Hand washing using soap after using the toilet at home *	Always	37,551	68.3	37,118	67.7	43.082	0.000
	Often	11,024	20.1	10,752	19.6		
	Sometimes	4891	8.9	5192	9.5		
	Not at all	1482	2.7	1786	3.3		
Returning home and hand washing using soap *	Always	39,115	71.2	38,993	71.1	44.276	0.000
	Often	9694	17.6	9260	16.9		
	Sometimes	4842	8.8	4994	9.1		
	Not at all	1297	2.4	1601	2.9		

Sample (N), percentage (%), chi-square test (*X*^2^), in the past week (*), *p*-value < 0.05.

**Table 5 healthcare-10-01881-t005:** Changes in daily life due to the COVID-19 outbreak.

Variable		Sample N	Percentage %
Physical activity	Large increase	3442	6.3
	Increase	7652	14.0
	No change	17,740	32.4
	Decrease	17,722	32.3
	Large decrease	8279	15.1
Not eating breakfast	Large increase	2354	4.3
	Increase	5558	10.1
	No change	39,791	72.6
	Decrease	4133	7.5
	Large decrease	2999	5.5
Drinking	Large increase	306	0.6
	Increase	1212	2.2
	No change	44,659	81.8
	Decrease	808	1.5
	Large decrease	7617	14.0
Smoking	Large increase	254	0.5
	Increase	273	0.5
	No change	45,078	83.6
	Decrease	466	0.9
	Large decrease	7864	14.6
Depression	Large increase	3758	6.9
	Increase	15,972	29.1
	No change	29,562	53.9
	Decrease	1937	3.5
	Large decrease	3606	6.6

## Data Availability

Not applicable.

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
