# Peer review of "Changes in Daily Life, Physical Activity, GAD, Depression, and Personal Hygiene of Adolescents in South Korea Due to the COVID-19"

_healthcare, 2022, doi:10.3390/healthcare10101881_

Round 1

Reviewer 1 Report

In the method, please add the inclusion and exclusion criteria of the subject.

Is there any publication regarding the Youth Health Behaviour Survey?

Author Response

Once again, we are grateful that the reviewer acknowledged the improvements in the manuscript, and we thank the editor and the reviewer for their additional constructive comments and criticisms. We have revised our manuscript as per the recommendations of the reviewers.

Point-by-point responses to your comments and changes made when revising the manuscript are provided below. The revised manuscript has been submitted to the website.

Here is a point-by-point response to the reviewer's comments and concerns.

We hope this modification meets your approval.

Reviewer 2 Report

The article is well structured, easy to understand and interesting. There are revisions to be implemented.

- in the abstract they should better specify the purpose of the study and the results obtained.

- authors should enhance the discussion entering and commenting the results

Author Response

Once again, we are grateful that the reviewer acknowledged the improvements in the manuscript, and we thank the editor and the reviewer for their additional very constructive comments and criticisms. We have revised our manuscript as per the recommendations of the reviewers.

Point-by-point responses to your comments and changes made when revising the manuscript are provided below. The revised manuscript has been submitted to the website.

Here is a point-by-point response to the reviewer's comments and concerns.

We hope this modification meets your approval.

Reviewer 3 Report

Dear authors,

The topic of the article is one of great interest and the approach is appropriate. I have indicated some changes that I consider useful to increase the value of the work. The most important of these concerns the reformulation of the Conclusions. 

So,

Line 19: Enter some additional details about the method used.

Lines 23-25: The use of quotations is not indicated in the Abstract. Remove the quotation marks and rephrase the sentence.

Line 29: Rephrase the text "and lifestyle are also restricted, confusion and difficulties", the meaning is not clear.

Lines 62-64: Rephrase the objectives of the article, it is not very clear what you want to find out through the research.

Table 4: Change the font on the Variable column

Discussion: Move the research output from Conclusions to the beginning of the Discussion chapter

Conclusions: Highlight the significance of the research findings. Show how they relate to the objectives of the article.

Move the paragraph about research limitations (lines 213-216) to the end of Conclusions.

In line 214 use "we" instead of "I".

With these changes, I believe that the article can be considered for publication. Success!

Author Response

Once again, we are grateful that the reviewer acknowledged the improvements in the manuscript, and we thank the editor and the reviewer for their additional constructive comments and criticisms. We have responded to the reviewers’ comments in detail below. All changes in the text of the manuscript are written in blue.

We hope this modification meets your approval.

Round 2

Reviewer 3 Report

Lines 250-255: I suggest to move the presentation of numerical data (percentages) also to the Discussions, and to keep in the Conclusions only formulations like "increased", "decreased", with references to the previous analysis.

Author Response

Once again, we are grateful that the reviewer acknowledged the improvements in the manuscript, and we thank the editor and the reviewer for their additional constructive comments and criticisms. We have revised our manuscript as per the recommendations of the reviewer.

Point-by-point responses to your comments and changes made when revising the manuscript are provided below. The revised manuscript has been submitted to the website.

Here is a point-by-point response to the reviewer's comments and concerns.

We hope this modification meets your approval.
